# Aquaporins Are One of the Critical Factors in the Disruption of the Skin Barrier in Inflammatory Skin Diseases

**DOI:** 10.3390/ijms23074020

**Published:** 2022-04-05

**Authors:** Paola Maura Tricarico, Donatella Mentino, Aurora De Marco, Cecilia Del Vecchio, Sabino Garra, Gerardo Cazzato, Caterina Foti, Sergio Crovella, Giuseppe Calamita

**Affiliations:** 1Institute for Maternal and Child Health IRCCS Burlo Garofolo, Via dell’Istria 65/1, 34137 Trieste, Italy; tricaricopa@gmail.com; 2Department of Biosciences, Biotechnologies and Biopharmaceutics, University of Bari “Aldo Moro”, Via E. Orabona, 4, 70125 Bari, Italy; donatella.mentino@uniba.it (D.M.); sabino.garra@uniba.it (S.G.); 3Section of Dermatology, Department of Biomedical Sciences and Human Oncology, University of Bari “Aldo Moro”, Piazza Giulio Cesare, 11, 70121 Bari, Italy; aurorademarco94@gmail.com; 4Dermatology Unit, Fondazione IRCCS Ca’ Granda Ospedale Maggiore Policlinico, 20122 Milan, Italy; ceciliadelvecchio2@gmail.com; 5Section of Pathology, Department of Emergency and Organ Transplantation (DETO), University of Bari “Aldo Moro”, Piazza Giulio Cesare 11, 70121 Bari, Italy; gerardo.cazzato@uniba.it; 6Section of Dermatology, Department of Biomedical Science and Human Oncology, University of Bari “Aldo Moro”, Piazza Giulio Cesare, 11, 70121 Bari, Italy; caterina.foti@uniba.it; 7Biological Science Program, Department of Biological and Environmental Sciences, College of Arts and Sciences, University of Qatar, Doha 2713, Qatar; sgrovella@qu.edu.qa

**Keywords:** membrane transport, aquaporin channels, AQP3, hidradenitis suppurativa, atopic dermatitis, psoriasis

## Abstract

The skin is the largest organ of the human body, serving as an effective mechanical barrier between the internal milieu and the external environment. The skin is widely considered the first-line defence of the body, with an essential function in rejecting pathogens and preventing mechanical, chemical, and physical damages. Keratinocytes are the predominant cells of the outer skin layer, the epidermis, which acts as a mechanical and water-permeability barrier. The epidermis is a permanently renewed tissue where undifferentiated keratinocytes located at the basal layer proliferate and migrate to the overlying layers. During this migration process, keratinocytes undertake a differentiation program known as keratinization process. Dysregulation of this differentiation process can result in a series of skin disorders. In this context, aquaporins (AQPs), a family of membrane channel proteins allowing the movement of water and small neutral solutes, are emerging as important players in skin physiology and skin diseases. Here, we review the role of AQPs in skin keratinization, hydration, keratinocytes proliferation, water retention, barrier repair, wound healing, and immune response activation. We also discuss the dysregulated involvement of AQPs in some common inflammatory dermatological diseases characterised by skin barrier disruption.

## 1. Introduction

The skin barrier provides an effective barrier between our body and the environment, preventing the entry of pathogens and protecting from chemical and physical attacks, as well as the unregulated loss of water and solutes [1,2,3].

In this context, aquaporins (AQPs), a family of membrane channel proteins that allow the osmotic movement of water and small neutral solutes, play an important role. AQPs are expressed in many different cell types of the skin and contribute to several of the skin’s key functions including hydration, formation of water-permeability barrier, and immune responses [4].

Alteration and defects of the skin barrier system at any level are often related to the development of variable skin disorders. Inflammatory skin diseases could be associated with skin barrier alterations, although the interplay between the inflammation and disruption of the skin barrier is complex and not fully unravelled yet; immunological dysfunction can cause barrier aberration or vice versa, and defects in the skin barrier functions—due, for example, to genetic mutations or mechanical stimuli such as scratching—can trigger inflammation responses [2].

Alterations in AQPs levels have been observed in inflammatory skin diseases with a defective skin barrier such as hidradenitis suppurativa, atopic dermatitis, and psoriasis [5,6,7,8,9,10,11,12].

In this review we focus on inflammatory skin diseases with a defective skin barrier, giving particular attention to the alterations of water and glycerol permeability with the involvement of aquaporin membrane channels. To this end, we will organize the knowledge on the expression, regulation, and complex roles and overall functions of most of the AQPs in the skin in health and in inflammatory skin disorders.

## 2. Skin Barrier and Water Permeability

The skin is widely considered the first line defence of the human body as it can display an essential role in rejecting pathogens and preventing mechanical, chemical, and physical damages. The skin barrier can be divided into three main functional levels: chemical barrier, physical barrier, and immune barrier [1,2,3].

Keratinocytes are the main actors in these barriers being responsible for the formation of the four layers underlying the epidermis, namely the stratum basale (SB), stratum spinosum (SS), stratum granulosum (SG), and stratum corneum (SC). The epidermis is a permanently renewed epithelium where undifferentiated keratinocytes located at the SB proliferate and migrate first to SS and then to SG. During this migration process, they undertake a differentiation program known as the keratinization process. Finally, these cells reach the most superficial layer (SC) composed of terminally differentiated keratinocytes, the corneocytes [13]. Corneocytes are protein-enriched, enucleated, and flattened keratinocytes cemented together by a dense lipid transcellular matrix [1,2].

The chemical barrier is mainly carried out by the SC thanks to its hygroscopic compounds called natural moisturizing factors (NMF) and epidermal lipids. NMF are found in corneocytes, and they are a mixture of amino acids and their derivatives, resulting from the proteolysis of epidermal filaggrin [3]. Instead, epidermal lipids are of sebaceous and keratinocyte origin and cover the surface of the skin filling the spaces among the cells [14] (Figure 1).

The nucleated epidermal cells, underlying the SC, contribute to the physical barrier function of the skin. Tight junctions and desmosomes, indeed, ensure epidermal cells’ cohesion, while collagen fibers and anchoring filaments create adhesion to the dermis, thus forming a solid system made of different elements constantly interacting with each other [2] (Figure 1).

The immune barrier consists of skin-resident immune cells populating the epidermis, such as keratinocytes, epidermal Langerhans cells, and the dermis, such as dendritic cells and resident T cells [3,15]. This barrier is crucial for the prevention of infection and for tissue homeostasis and reconstruction [16] (Figure 1).

The chemical and physical barriers together are important for maintaining the skin water content. In this context, the epidermal lipids, which form a hydrophobic layer, together with the tight junctions and desmosomes prevent dehydration and retain water inside the skin. It is interesting to note that the skin water content is different among the various layers by being high in SB since keratinocytes need water to proliferate and differentiate, and drastically decreasing in SC since corneocytes constitute the low water content permeability barrier [17].

However, barrier-to-water permeation is not absolute as a basal water movement through the skin, which normally occurs under the name of transepidermal water loss (TEWL) [1].

An inverse relationship between TEWL and skin hydration has been well demonstrated, as higher levels of TEWL, a marker of altered skin barrier function, are frequently correlated with a lower water content in the SC [2].

Alteration and defects of the skin barrier system at any level are often related to the development of variable skin disorders, which may be congenital, when the defect is genetically determined, or acquired, when the defect occurs later on in a patient’s life [18].

## 3. Aquaporins

Aquaporins (AQPs) are a family of membrane channel proteins widespread in nature that facilitate the bidirectional transport of water and small neutral solutes across biological membranes [19,20]. Mammals express thirteen distinct AQPs (named AQP0-12) that are grossly classified into three main groups based on their pore permeability: (i) *orthodox aquaporins* (AQP0, AQP1, AQP2, AQP4, AQP5, AQP6, and AQP8), homologues that were initially believed to transport only water, (ii) *aquaglyceroporins* (AQP3, AQP7, AQP9, and AQP10), AQPs allowing the permeation of small solutes, particularly glycerol, in addition to water, and (iii) *superaquaporins* or *unorthodox aquaporins* (AQP11 and AQP12), isoforms whose permeability remains to be fully assessed and that are marked by a distinct evolutionary pathway. However, this subdivision is not fully exhaustive since additional solutes can cross the pore, sometimes even more efficiently than the molecules that were initially considered to be the only substrates. AQP3, AQP6, AQP8, and AQP9 can also permeate ammonia, the reason why they are also termed *ammoniaporins* or *aquaammoniaporins* [21]. AQP1, AQP3, AQP5, AQP8, AQP9, and, as recently reported [22], also AQP11, allow the movement of hydrogen peroxide; hence, they are also counted as *peroxiporins* [23,24]. Moreover, AQP7 and AQP9 have been reported to mediate the transmembrane transport of arsenite, AQP6 to transport nitrate/halide ions, and AQP9 to facilitate monocarboxylate transport [25]. Some AQP channels have also been reported to mediate the transmembrane exchange of physiologically important gases, such as CO_2_, NO, or O_2_ [26,27,28], although these gases can freely diffuse through the phospholipid bilayer. Expression, transport properties, and chemical gating [29,30] of AQPs are matters of intense investigation in all body districts, and several important roles, some of them even unexpected [31,32], have been described in health and disease (for review, see [19]). Their involvement in a number of diseases, including those affecting the skin, makes these protein channels appealing targets for the development of new drugs [29,33,34].

## 4. Expression, Regulation, and Physiological Relevance of Skin Aquaporins

At least six distinct AQPs (AQP1, 3, 5, 7, 9, and 10) are selectively expressed in various skin cell types (Table 1).

From deep to superficial, AQP7 is located primarily in the hypodermis (but also in dermal and epidermal dendritic cells [35]), AQP5 is found in the eccrine sweat glands [36], and AQP9 and 10 are located in the epidermis. AQP1 and AQP3 have been found in both the dermis and epidermis. This review will discuss the expression, regulation, complex roles, and overall functions of most of these AQPs in the skin in health and in inflammatory disorders.

### 4.1. AQP1

AQP1 is found in dermal fibroblasts and vascular endothelial cells and has also been detected in melanocytes, located in the SB of the epidermis [37]. The main function of AQP1 is played in the vascular endothelial cells, where it mediates the exchange of water between the blood and dermis to maintain hydration. Less well elucidated is the physiological meaning in fibroblasts and melanocytes. In fibroblasts, AQP1 has been found to be upregulated during periods of hypertonic stress [38]. A similar increase in AQP1 expression has been postulated in melanocytes during conditions of osmotic stress; however, further work is needed [39]. AQP1-mediated water transport has also been suggested to occur in keratinocyte migration. Cell migration was restored in keratinocytes derived from AQP3-deficient mice by infecting the AQP1-depleted cells with either AQP3 or AQP1 [40]. The influx of water through either AQP1 or AQP3 was speculated to provide the hydraulic pressure needed to extend processes for cell movement [40].

### 4.2. AQP3

AQP3 is expressed in keratinocytes from the SB to the SG of the epidermis where it is localized at the plasma membrane (Figure 2).

At a much lower extent than the plasma membrane level, AQP3 is also detected in the intracellular compartment of keratinocytes (Figure 2). This pattern is consistent with an in vitro study reporting the strong downregulation of AQP3 mRNA levels as basal keratinocytes reach late differentiation [41]. Plasma membrane AQP3 has been reported to be associated with phospholipase D2 (PLD2) in caveolin-rich membrane microdomains, leading to the hypothesis that glycerol imported through AQP3 is used by PLD2 in the transphosphatidylation reaction that generates phosphatidylglycerol (PG), a phospholipid that acts as a signalling molecule to mediate early epidermal keratinocyte differentiation. Manipulation of this signalling module was reported to block keratinocyte proliferation and increase differentiation [42]. The mechanisms through which the AQP3 expression is regulated in the epidermis are just starting to be understood. A recent study with keratinocytes and mouse skin ex vivo reported that histone deacetylase-3 (HDAC3) has a role in modulating AQP3 expression. The levels of AQP3 mRNA and protein were found as increased after the selective inhibition of HDAC3, suggesting that in epidermal keratinocytes under basal conditions, HDAC3 suppresses AQP3 expression [43]. A role for p53 transcription factors in increasing AQP3 has been also suggested [43]. However, neither the possibility that the enhanced AQP3 expression in response to HDAC inhibition results from a generalized increase in gene transcription by means of effects on chromatin structure, through the stimulation of the activity of one or more transcription factors including p53 or another family member, nor a combination of these processes can be ruled out. The AQP3 promoter contains a p53 response element that can also be bound and activated by p73 [44]. The activation of PPARγ has been shown to lead to a marked increase of the AQP3 mRNA in both undifferentiated and differentiated cultured human keratinocytes (CHKs). The increase in AQP3 transcript by PPARγ agonists occurred in a dose- and time-dependent manner. The increase in AQP3 mRNA level was followed by an augmentation in AQP3 protein in undifferentiated keratinocytes and a considerable increase in glycerol uptake [45]. The topical application of ciglitazone, a PPARγ activator, also increased AQP3 expression in mouse skin in vivo. Similar effects on AQP3 expression were seen after activating other nuclear hormone receptors such as liver X receptors (LXR), retinoic acid receptors (RAR), and retinoid X receptors (RXR) [45]. Consistent with this observation, Yang and colleagues reported that PPARs contribute to the HDAC inhibitor-induced increase in AQP3 levels as PPAR antagonists were seen to prevent the HDAC inhibitor-dependent elevation of AQP3 levels. PPARγ upregulation was accompanied by the HDAC inhibition-stimulated AQP3 mRNA and protein expression [46]. However, further work is needed to evaluate whether the inhibition of HDAC leads to an increase in the accessibility of the gene promoter of AQP3 or modifies the acetylation of PPAR to stimulate its transcriptional activity. AQP3 possesses a N-linked glycosylation site at loop C, and in several cell types, this consensus seems to be functional, as besides the core protein of about 28 kDa, glycosylated forms with molecular masses ranging between 40 and 60 kDa are also present. AQP3 appears to be glycosylated also in keratinocytes [47] where, as for other AQP homologues, proper glycosylation may affect the function and subcellular localization of the protein [48]. In murine epidermal keratinocytes, AQP3 was downregulated by differentiating agents such as 1,25-dihydroxyvitamin D3 and high levels of extracellular calcium (1 mM) [41]. This was accompanied by a reduction of keratinocyte glycerol uptake. However, glycerol uptake is increased by moderate extracellular calcium (125 µM) [49] at a level known to promote keratinocyte differentiation in vitro [50]. Qin and coworkers reported in vitro evidence that 125 µM extracellular calcium decreases the non-glycosylated form of AQP3 and increases the glycosylated form of the protein [42]. If glycosylation favours the plasma membrane localization as seen for AQP2 in kidney cells [51], this fits with the reported pattern of localization of AQP3 in the suprabasal skin layers of human [52] and mouse [42] skin. This possibility does not seem to fit with a study by another group using cultured human keratinocytes where no effect of elevated Ca^2+^ concentration on AQP3 protein levels was observed [45]. However, in the same work, it was shown that the effect of PPAR agonists in increasing AQP3 was limited to the glycosylated form of the protein, suggesting that, also in human keratinocytes, differentiating agents increase glycosylated AQP3 levels. Phosphorylation may be another post-translational modification regulating the subcellular localization of AQP3 as occurs for AQP2 in the principal cells of renal-collecting ducts’ translocation to the plasma membrane. However, no evidence exists supporting this eventuality. In keratinocytes, AQP3 has been reported to translocate to the plasma membrane upon exposure to osmotic stress [53]. A final post-translational modification that may affect it is protein acetylation. Renal AQP3 was shown to be acetylated on lysine 282 [54], and a similar modification may occur in keratinocyte AQP3, influencing its subcellular localization and/or function. Hence, the promotion of lysine acetylation by HDAC inhibition might exert actions on AQP3 in addition to its effects on expression. Mechanisms other than post-translational modifications may also regulate AQP3. A basolateral targeting sequence (YLLR) is found in AQP3 from multiple species [55]. Although the current literature provides interesting information about the regulation of AQP3 in the skin, additional studies are needed to fully assess the posttranslational modifications of AQP3 and the role of these modifications on the subcellular localization and function of this channel.

Key roles have been ascribed to AQP3 in various processes underlying keratinocyte function. The importance of AQP3 in skin is also indicated by the dysfunctions observed in a number of human skin diseases. A role for AQP3 is often suggested in keratinocyte differentiation. By an in vitro study, Bollag and colleagues suggested a function of AQP3 in early mouse keratinocytes differentiation [56]. AQP3-mediated import of glycerol inhibited cultured keratinocytes proliferation, leading to the appealing hypothesis that AQP3 is pivotal in arresting the growth of basal keratinocytes after they move into the first differentiated layer of the SS. This possibility was in line with a subsequent study with non-melanoma skin cancer cells characterised by excessive proliferation. AQP3 was greatly decreased in the lesions compared to the overlying normal-appearing epidermis [57]. A role for AQP3 in promoting keratinocyte differentiation was also proposed after co-expressing AQP3 with constructs in which the promoters of keratinocyte differentiation markers drove the expression of a reporter enzyme [56] and in a study where re-expression of the protein in AQP3-knockout keratinocytes led to increase of the mRNA and protein levels of differentiation markers either alone and/or in combination with an elevated concentration of Ca^2+^ [47]. Knockdown of AQP3 inhibited the expression of keratin 10 in keratinocytes exposed to high Ca^2+^ levels to induce differentiation [58]. The same study also suggested a role for AQP3 in promoting keratinocyte viability. The association between keratinocytes’ AQP3 levels and early differentiation through Notch1 signalling was reported by Guo and colleagues [10]. In spite of the above evidence on the role of AQP3 in keratinocytes, differentiation remains rather controversial. After knocking down AQP3, Verkman and coworkers found no changes in the levels of differentiation markers in human keratinocytes induced to differentiate by high Ca^2+^ concentration [59]. Moreover, no changes in the basal expression of keratinocyte differentiation markers were seen in the epidermis of knockout mice lacking AQP3. Based on their previous studies indicating a physical and functional association between AQP3 and PLD2, Bollag and colleagues proposed that the ability of AQP3 to induce keratinocyte differentiation depends on its interaction with this lipid-metabolising enzyme phospholipase [60]. As mentioned above, these authors suggested that PLD2 converts the AQP3-transported glycerol to PG via transphosphatidylation, promoting keratinocyte differentiation through the AQP3/PLD2/PG signalling pathway [56]. However, further work is needed to unravel the apparent discrepancies regarding the role of AQP3 as a glycerol channel in skin keratinocyte differentiation. It should also be considered that glycerol can cross the plasma membrane by moving by simple diffusion through the phospholipid bilayer [61].

An important role for AQP3 has been suggested in keratinocyte proliferation by a series of studies conducted by Verkman and colleagues [8,40,59,62,63]. Work with AQP3 knockout mice indicated that this role is not played under basal conditions in vivo since the mice with the ablation of AQP3 showed similar epidermal thicknesses and layer numbers as well as proliferation compared with wild-type mice [62]. In a subsequent study, the *Aqp3*^−/−^ mice showed reduced tumour formation when submitted to topical application of a carcinogen followed by treatment with a tumour promoter [63]. Moreover, after tumour promoter activation, the *Aqp3* knockout mice showed less epidermal thickening and a smaller increase in the number of proliferating cells compared with the wild-type mice. The reduced keratinocyte proliferation was interpreted as due to decreased cellular ATP levels caused by the lack of AQP3 and the consequent reduced cellular uptake and metabolism of glycerol. Indeed, reduced epidermal glycerol content was found in AQP3-deficient mice [62], and supplementation with glycerol corrected the defect in keratinocyte proliferation induced by wounding [40,63]. In a study with normal human keratinocytes AQP3 upregulation was also seen to increase glycerol uptake, keratin 5 and 14 expression, and cell growth, whereas AQP3 silencing inhibited proliferation in response to the cytokine CCL17 [8]. A similar result was seen in vivo in AQP3-ablated mice challenged with retinoic acid [59]. Comparable results were obtained by Guo and coworkers who found that AQP3 knockdown diminished human keratinocyte proliferation and increased the expression of early (keratin 10), intermediate (involucrin), and late (filaggrin) differentiation markers [10]. Roles for AQP3 in promoting cell proliferation have also been reported in several cancers (for review see [64]). AQP3 has also been found to promote epithelial-mesenchymal transition in gastric cancer cells [65] and to facilitate epidermal cell migration during wound healing. Knockdown of AQP3 in normal human keratinocytes decreased glycerol uptake, scratch wound healing, and fetal bovine serum-induced migration in vitro [40]. Keratinocytes isolated from AQP3-deficient mice showed reduced migration compared with cells originating from wild-type mice [40]. Adenoviral-mediated expression of either AQP1, an orthodox aquaporin water channel with no permeability to glycerol, or AQP3 was able to restore normal epidermal cell migration [40]. This suggests that the effect of AQP3 in skin keratinocyte migration relates to the facilitation of water movement rather than the transport of glycerol. However, the facilitation of hydrogen peroxide uptake cannot be ruled out since AQP3 has good peroxiporin activity, and convincing evidence exists indicating relevance of AQP3-mediated H_2_O_2_ import in cell migration [66,67].

As anticipated above, AQP3 also exerts an important function in skin wound healing, a role in line with its involvement in cell proliferation and migration. AQP3-ablated mice showed delayed wound healing of full-thickness skin wounds accompanied by decreased keratinocyte proliferation, and this impairment was rescued through glycerol supplementation [40]. Consistent with this result, AQP3 expression was decreased in the wounds of diabetic rats with impaired wound healing [68]. However, whether AQP3 is also involved in human skin wound healing needs to be proved since rodents are not ideal models for human skin wound healing (rodent wounds heal mainly by contraction, whereas human wounds heal primarily by re-epithelialization). Moreover, wound healing is an extremely complex process that involves many other cell types, thus making it even more difficult to assess the exact role of AQP3 in its development. Besides keratinocytes, indeed, endothelial cells and myofibroblasts represent the major cellular actors in the wound healing process, especially regarding inflammation and wound contraction, respectively [69]. Therefore, if on one hand endothelial cells take part to the initial cellular crosstalk that promotes local inflammation, vasodilation, and haemostasis, thus facilitating tissue repair and growth factors concentration, myofibroblasts are required at the end of the process in order to reconstitute an efficient extracellular matrix as well as to restore tissue deficiency when re-epithelialization is not possible or sufficient [70,71]. Nevertheless, due to the central role of re-epithelialization in human wound healing and the relevance of AQP3 in promoting the proliferation and migration of human keratinocytes, it is reasonable to hypothesize a consistent role of AQP3 in human skin wound healing as well. Convincing evidence exists indicating an important role of AQP3 in skin hydration. AQP3-deficient mice showed reduced SC hydration compared to the wild-type control mice [62,72], and the difference between control and *Aqp3*^−/−^ knockout mice was abolished after the exposure of the mice to low (10%) humidity [72]. No differences were seen in SC morphology, thickness, lipid content, or levels of a variety of metabolites, whereas a decrease in SC and epidermal glycerol content was observed [72]. The SC hydration defect was corrected by restoring the epidermal glycerol levels [73]. The pivotal role of AQP3 in skin hydration is also indicated by its circadian rhythm of expression, and skin hydration correlates with these cyclical AQP3 levels [74]. Glycerol generation from triglyceride in sebaceous glands is fundamental for SC hydration, as also indicated by a study with the asebia mouse model where defect in sebum production leads to a reduction in epidermal glycerol content and abnormal SC hydration, which in turn leads to hyperkeratosis (epidermal thickening), epidermal hyperplasia, and mast cell activation. This phenotype can be resolved by the topical administration of glycerol but not urea or water [75], suggesting a key physiological role in skin function and the involvement of AQP3 in preserving epidermal glycerol content and SC water-holding capacity. In a mouse model of streptozotocin (STZ)-induced diabetes, changes in epidermal AQP3 expression have been invoked to explain the skin dryness (xeroderma) accompanying diabetes, and the disruption of the circadian rhythm was suggested as a possible mechanism through which STZ-induced diabetes contributes to AQP3 downregulation [76].

AQP3 has also been reported to be important in the function of the epidermal water permeability barrier. Although basal skin barrier function was not impaired, AQP3-ablated mice showed an approximately two-fold delay in the repair of their water permeability barrier compared with the wild-type control as measured by transepidermal water loss after tape stripping [62]. Again, as occurred for other altered skin functions, oral glycerol administration restored the impaired barrier recovery of AQP3-deficient mice and even accelerated recovery in the wild-type control [62]. In line with these results, Bollag and colleagues found an accelerated water permeability barrier repair in a transgenic mouse where AQP3 was selectively overexpressed in the epidermis under the control of the keratin 1 promoter [42].

AQP3 is also expressed in dermal-resident T cells, where it was suggested to regulate their trafficking in cutaneous immune reactions after observing that AQP3-ablated mice are defective in the development of hapten-induced contact hypersensitivity [67]. The impaired trafficking of antigen-primed T cells to the hapten-challenged skin was attributed to the AQP3-mediated H_2_O_2_ uptake required for the chemokine-dependent migration of T cells.

### 4.3. AQP5

Eccrine sweat glands are composed of single tubular structures containing acinar cells and ductal cells. Mouse, rat, and human eccrine sweat gland secretory and excretory cells express AQP5 that, upon stimulation, traffics at their apical membrane [36,77,78]. Acinar cells secrete a primary fluid rich in ions and water whose composition is modified by reabsorption by the ductal cells [79]. The role of AQP5 in eccrine sweat glands remains an open debate due to variable data acquired with different AQP5 knockout mice strains and methods to assess the secretion [36,80]. Zhou and coworkers are the unique authors detecting AQP5 in keratinocytes where they suggested a role for this AQP in the balance between epidermal keratinocyte proliferation and differentiation [81]. However, further studies are needed to evaluate if the expression of AQP5 in keratinocytes in vivo is physiologically relevant and, eventually, clarify its role in the epidermis.

The presence of AQP5 in human axillary sweat glands has been reported, and upregulation of this orthodox AQP was detected in axillary sweat glands of primary focal hyperhidrosis (PFH) patients. PFH is a sweat disorder characterised by excessive sweating in specific body areas, such as the palms, axillae, feet, or forehead, where the eccrine sweat glands are concentrated [82]. This is consistent with a work by Ma and coworkers who used a mouse model and reported that topiramate reduced sweat secretion along with a diminished AQP5 expression by sweat glands, suggesting that AQP5 may be involved in topiramate-induced hypohidrosis [83]. These observations conflicted with a previous study where no significant difference in AQP5 expression was observed in the palmoplantar skin of healthy subjects and patients with a form of palmoplantar hyperhidrosis [84]. Thus, the main role of AQP5 in human sweat glands remains to be fully clarified; further investigation is needed to increase the understanding of the pathogenic mechanisms of hyperhydrosis and the hypothesized implication of AQP5.

### 4.4. AQP7

Dendritic cells (DCs) feature the ability to present antigen and exert a critical role in the induction of the acquired immune response. The immune system of the skin relies on a rich network of antigen-presenting DCs that localise in the epidermis and the dermis. Skin DCs are distinct into three subsets, epidermal Langerhans cells (LCs) and Langerin- or Langerin+ dermal DCs (dDCs). LCs and dDCs uptake antigen and subsequently migrate to regional draining lymph nodes, where they activate naive T cells by initiating the immune response. A study using AQP7-deficient mice and isolated LCs and dDCs from mouse skin found that AQP7 is functionally expressed in mouse LCs and dDCs and plays roles in antigen uptake, cell migration, and the subsequent initiation of an immune reaction [35]. AQP7 was suggested to play an important role in allergy induction and immune surveillance in the skin and, likely, in other tissues in which DCs are localized.

### 4.5. AQP9

AQP9 is a homologue of AQP3 with broad selectivity and is predominantly expressed in leukocytes [85] and the liver [61]. AQP9 was detected in the upper SG of the human and mouse epidermis [86]. AQP9 knockout mice did not show any apparent defect, including wound healing [86]. AQP9 mRNA expression was only detected in cultured, differentiated normal human epidermal keratinocytes [87]. Stimulation with retinoic acid, a potent stimulator of keratinocyte differentiation, increased AQP3 expression and downregulated AQP9 expression, suggesting that AQP9 expression in epidermal keratinocytes undergoes a different regulation compared to that of AQP3 [88]. The treatment of cultured keratinocytes with the keratinocyte differentiating agent VD3 also significantly increased AQP9 expression, an observation that has led to the hypothesis that AQP9 plays a role in highly differentiated keratinocytes. In human epidermis, AQP9 and occludin, a tight junction marker, co-localize in the upper stratum granulosum [88]. Considering that tight junctions function as a paracellular barrier against small molecules, besides having a role in the terminal differentiation of keratinocytes, another function for AQP9 in human epidermis may be that of constituting a transcellular route for the movement of solutes of biological relevance, such as glycerol and urea, into and out of the skin.

The involvement of neutrophil AQP9 in hapten-induced contact hypersensitivity has been shown in a murine model of skin allergic-contact dermatitis using *Aqp9*^−/−^ knockout mice [89]. AQP9 was therefore suggested to be required for the development of sensitization during cutaneous acquired immune responses via regulating neutrophil function.

### 4.6. AQP10

AQP10 is another aquaglyceroporin detected in the epidermal keratinocytes. By in vivo studies, AQP10 has been localized to the SC of the human epidermis [90]. AQP10 is believed to share similar meaning as AQP3, as it may play a role in the barrier function of the skin due to its expression in the SC [39,90]. The involvement of AQP10 in the pompholyx has also been suggested [91], although further studies are needed to precisely address this possibility.

## 5. Skin Aquaporins in Inflammatory Dermatological Diseases

Besides its role as a mechanical barrier restricting water loss and preventing dangerous substances and microorganisms to penetrate, human skin also represents the first line immunological defence against infections. A complex crosstalk among all skin cell types regulates immune responses in the skin to ensure an effective host defence and to preserve tissue integrity. Moreover, skin microbiota, including bacteria and fungi, represents a strong deterrent for dangerous microbial overgrowth, also influencing local immune responses through the interaction with both epithelial and immune cells. For these reasons, skin immune homeostasis relies upon a delicate balance in which many different cell types take constant part. On the other hand, the dysregulation of this fragile equilibrium leads to the development of different skin inflammatory disorders such as atopic dermatitis, psoriasis, and even hidradenitis suppurativa [92].

Besides their physiological role and modulation in well-balanced inflammatory responses, AQPs have been found to be dysregulated during several inflammation-based diseases [93]. The alteration of skin AQP expression has been observed in the above-mentioned three common inflammatory dermatological diseases: hidradenitis suppurativa, atopic dermatitis, and psoriasis (Figure 3)

### 5.1. Hidradenitis Suppurativa

Hidradenitis suppurativa (HS), also known as *Acne Inversa*, is an inflammatory skin disorder affecting preferably women (female:male ratio, 3:1) in their second and third decades of life [94]. Patients generally develop inflamed nodules and abscesses that can evolve into epithelialized transcutaneous tunnels (“sinus tracts”) and disfiguring scars, mainly occurring on intertriginous areas [95].

The most commonly used staging system for HS in the clinical practice worldwide is Hurley staging, which classifies the disease into three levels of severity, depending on the number and on the conformation of skin lesions, including nodules, abscesses, sinus tracts, and scars [96].

Although its pathogenesis is still unclear, many risk factors have been identified, including genetic predisposition, obesity, smoking habit, and hormonal disorders [97].

Moreover, recent studies suggest that HS may be a disorder of the follicular epithelium rather than a disease affecting the sweat glands, opening new interesting scenarios regarding its aetiology. However, despite its pathogenetic mechanisms being unknown, it is now certain that HS is an autoinflammatory disease, as the infective theories are now almost completely abandoned. As a matter of fact, superinfection of the most common lesions may occur at any time but only as a secondary process, thus explaining the double role of antimicrobial therapy in HS as both the prevention and treatment of eventual superinfections [94,98]. An aberrant immune response is crucial in the development of the disease. HS can be considered a Th1/Th17-driven inflammatory skin disease, as demonstrated by the histological examination of skin biopsies, revealing high concentrations of Th17 in lesional HS skin. Many cytokines are indeed involved in this inflammatory process, including IL-17, IL-23, IL-1β, TNF-α, and IL-12. While neutrophils and Th17 are the major sources of IL-17, other cell types such as keratinocytes release many other pro-inflammatory molecules, thus aggravating skin inflammation [99].

In 40% of cases, HS may occur as a familial disorder [100]; affected families with an autosomal dominant mode of inheritance and incomplete penetrance have been identified [101,102]. Mutations in genes encoding for three of the four proteins of g-secretase have been identified as the most common genetic variants involved in HS familiar cases. These genetic variations result in a dysfunction of the g-secretase activity that induces alterations in Notch signalling; alterations of Notch signalling can induce a failure in the regulation of keratinocytes functions, in particular in keratinocytes proliferation and differentiation [103].

Unfortunately, information on skin hydration and the TEWL of HS patients is currently insufficient. To our knowledge, only Bhargava and coworkers examined these parameters in various dermatological patients, including HS patients; they observed a significant increase in TEWL and decrease in skin hydration in inguinal skin [104].

Recent skin transcriptome analyses, conducted by Coates and coworkers, reported a downregulation of the AQP5 gene in HS lesions. In detail, in order to identify differential expressed genes, they analysed a publicly available microarray dataset of the lesional and non-lesional skin of HS patients, also compared to an RNA-seq dataset of wounded and non-wounded human skin samples, assuming that HS lesional skin may resemble a chronic non-healing wound. These data revealed the upregulation of AQP5 in wounded skin and downregulation in HS skin; the differential expression of AQP5 may suggest a role of this gene in HS pathogenesis [5].

As mentioned above, AQP5 is mainly distributed in the sweat gland secretory cells and sweat gland excretory duct cells. Nejsum et al. established an essential role for AQP5 in sweat production; in fact, in the *Aqp9*^−/−^ knockout mice model they observed an impairment of sweat secretion [36]. In addition, AQP5 can regulate the balance between epidermal keratinocyte proliferation and differentiation; in fact, AQP5 overexpression in a keratinocyte cell line (HaCaT cells) could induce proliferation and dedifferentiation [81].

The integration of all these results could lead us to conclude that the decrease in the AQP5 expression in the eccrine glands of HS lesional skin can compromise sweat generation and can also interfere with normal keratinocyte turnover, in particular in proliferation and differentiation.

### 5.2. Atopic Dermatitis

Atopic dermatitis (AD) is a chronic-relapsing inflammatory skin condition with variable clinical manifestations, depending on its activity phase [105,106]. AD is characterised by a strict differentiation of CD4 lymphocytes towards the Th2 lineage, thus leading to the increased production of IL-4, IL-5, and IL-13 within the skin. Interleukins and cytokines stimulate IgE antibodies and eosinophils in peripheral blood and tissues, while skin inflammation becomes responsible for the damage of the epidermal barrier, overlapping with the primary intrinsic defects of the barrier [107].

Acute AD, most common in early childhood, generally occurs as a vesicular, oedematous, erythematous, and crusting eruption, while subacute AD is characterised by dry, scaly and erythematous patches and chronic AD by very pruritic thickened lesions [105]. The most commonly affected sites are the flexural surfaces of both upper and lower limbs, cheeks, eyelids, neck, and wrists, with possible differences according to patients’ ages [105].

AD pathogenesis is complex and multifactorial, thus engaging a combination of genetic variations, epidermal barrier dysfunctions, and altered immune responses [108].

Numerous gene variations have been associated with a predisposition to AD, many of which encode for structural and functional proteins of the epidermis. Among these, variations in the filaggrin gene have been identified as the most common genetic variations involved in AD-familiar cases; these alterations are linked to the impairment of the skin barrier, in particular in the NMF [109].

Epidermal barrier dysfunctions in the lesional and nonlesional skin of patients is a critical feature of AD; these dysfunctions result in a greater transepidermal water loss (TEWL) and also a decrease in skin hydration [110,111].

As it occurs in HS, also in AD Notch signalling, a critical pathway regulating keratinocyte differentiation and proliferation seems to play a role; in fact, all Notch receptors are downregulated in the epidermis of the lesional skin of AD [103,112].

In recent years, three different AQPs have been reported to be involved in AD, in particular AQP5, AQP9, and AQP3.

Tian and coworkers observed a downregulation of AQP5 expression in lesions of AD patients with allergic rhinitis. In detail, in order to identify differential expressed genes, RNA-seq was conducted on lesional and nonlesional skin samples from 10 AD patients, matched with normal skin samples from 5 healthy volunteers. The downregulation of AQP5 was confirmed by qRT-PCR and Western blotting results [6].

As mentioned above, AQP5 is mainly distributed in sweat gland secretory cells and sweat gland duct cells and can regulate the balance between epidermal keratinocyte proliferation and differentiation [36,81]. As it has been observed in HS (previous paragraph), there is also a downregulation of AQP5 expression in AD. Unfortunately, to date there are no hypotheses to explain its role in AD, and it must also be considered that AQP5 alteration has only been observed in one study, in a subtype of AD, so it may be specific only for this particular subtype [6].

The downregulation of AQP9 was observed by Ghosh and coworkers analysing publicly accessible AD microarray datasets from five independent studies [7].

In the epidermis, AQP9 is expressed in the upper layer of the SG [88]. Rojek et al., in a AQP9 knockout mouse model, observed no apparent defects in wound healing [86]. To date, there is little information on AQP9 and skin.

Instead, several studies have demonstrated an involvement of AQP3 in AD. Nakahigashi et al. found a significant increase in AQP3 expression on keratinocyte membranes in AD skin lesions compared to a healthy control epidermis [8], while both AQP3 mRNA and protein expressions have been detected upregulated in the skin of AD murine models by Sun et al. [9].

Guo et al. showed a novel AQP3-Notch1 axis important in epidermal homeostasis. They observed that the elevated expression of AQP3 reduces the expression of filaggrin and increases the expression of other inflammatory markers; these AQP3-induced changes are normally observed in AD, in which Notch is inhibited [10].

One of the histological features associated with AD is an enhanced keratinocyte proliferation and impaired differentiation; possible causes can also include AQP3 altered expression levels, since these membrane channels promote cell growth and proliferation through an increase of the cellular uptake of glycerol. Furthermore, the exceeding cell proliferation leads to a damaged barrier function, another typical aspect of this skin disease [8].

Given these findings, upregulated AQP3 in AD lesions probably contributes to the formation of a defective skin barrier, due to its ability to support keratinocytes’ abnormal proliferation and differentiation; the non-functioning epidermal barrier then results in a greater skin water loss and an active immune response.

### 5.3. Psoriasis

Psoriasis (PS) is a chronic inflammatory skin disease primarily caused by both an altered skin barrier function and a dysregulated skin immune response. Innate immune cells indeed initiate psoriatic inflammation by producing many different proinflammatory cytokines such as IL-17 and IL-22, which directly interact with keratinocytes and other immune cells, thus inducing local inflammation and keratinocytes proliferation. In more than 90% of cases, PS is clinically characterised by sharply demarcated, erythematous plaques covered in silvery scales, generally occurring on the knees, elbows, scalp, and lumbosacral region [113,114,115]. Other clinical presentations may include flexural PS, guttate PS, and even more severe forms such as pustular PS and erythrodermic PS [113,114].

Furthermore, PS may represent a serious threat to patients’ global health as it is quite frequently associated with severe comorbidities such as cardiovascular diseases, diabetes, metabolic syndrome, and arthritis [113,114].

The major players of PS pathogenesis are keratinocytes and T-cells that cause alterations in the skin barrier; in fact, the lesions show keratinocytes hyperproliferation and the unregulated infiltration of T-cells [116].

A role of Notch signalling in PS pathogenesis has also been highlighted, although it is controversial; both the upregulation and downregulation of Notch pathway components have been investigated in psoriatic skin lesions, connected to hyperproliferation and altered differentiation as well as to vascular dysfunction [103].

Keratinocytes are now known to be involved in every step of PS development, from the early stage, when they initiate the inflammation process via chemokines and antimicrobial peptides, to the pathology maintenance step, when the crosstalk with IL-17-producing T-cells promote chronic inflammation and epidermal hyperplasia [117].

The inflammatory environment, the excessive keratinocytes proliferation, and dysregulated keratinization the prevent skin of a psoriatic lesion to exert its barrier function, resulting in increased TEWL and decreased skin hydration [118].

Lee and colleagues compared the skin of psoriatic patients with a healthy control, and they identified a reduction in AQP3 protein expression in lesional and perilesional skin, together with increased TEWL and decreased skin hydration. The cause of AQP3 protein loss on lesional keratinocytes is not yet known, although previous studies have found that some inflammatory soluble factors present in psoriatic skin, such as TNF-alpha, can inhibit AQP3 expression [11].

A different role of AQP3 in PS pathogenesis was found by Hara-Chikuma and colleagues. In a study with AQP3-deficient mice and AQP3-deficient keratinocytes, they connected NF-kB pathway activation with AQP3 channel activity through extracellular H_2_0_2_ uptake. H_2_0_2_ is a reactive oxygen species (ROS), and it is transported inside keratinocytes in response to TNF-alpha, where it contributes to NF-kB activation and the consequent regulation of inflammatory responses. AQP3 was reported as the channel through which H_2_0_2_ enters the keratinocytes. The AQP3-ablated mice models showed notable reduction in PS development compared with wild-type PS models, and the AQP3-deficient keratinocytes showed a decrease of H_2_0_2_ intracellular uptake and, therefore, a diminution of NF-kB-activated inflammation responses [12]. A similar mechanism involving NF-κB signalling in the inflammatory response has been recently reported for another peroxiporin, AQP9, in the liver [119].

Regardless of the conflicting results, it is possible to underline a role of AQP3 activity in psoriasis development and progression. The inflammatory environment seems to lower this channel expression in lesional skin, resulting in impaired water loss and hydration as well as dysfunctions in keratinocytes proliferation and differentiation. Moreover, AQP3 appears to promote inflammatory responses through the activation of the NF-kB pathway.

## 6. Summary and Conclusions

The role of AQP3, AQP5, and, to a certain extent, AQP9 has been disclosed by several authors in different independent experimental settings. Now, the question is, how can we use this information in a clinical setting? Can the quantitative evaluations of AQP3, AQP5, and AQP9 be envisaged for the severe forms of HS, AD, and PS?

All in all, based on the still-growing evidence shown in this review, we can affirm that AQPs are involved in skin physiology, with dysregulated expression in several skin disorders. Moreover, AQPs have been shown to be key players contributing to skin barrier disruption in inflammatory skin diseases. The mechanisms through which AQPs are modulated in skin physiology, particularly in relation to their involvement in skin’s mechanical and water-permeability barrier, wound healing and immune system response are still matters of debate with several experiments, including integrated OMICs analysis, which is still needed to enhance the knowledge on AQPs related mechanisms in skin diseases. Additional roles for AQPs in the skin are sure to be unravelled in the next years, especially if we consider that these membrane channels are also permeable to a series of other molecules, starting with a molecule that strongly impacts intracellular signalling such as hydrogen peroxide, in addition to water and glycerol. The existing discrepancies regarding the role of AQPs in skin keratinization may arise from attempting to translate the information obtained with cellular and animal models to human beings. The pathophysiological involvement of AQPs in the onset of human skin diseases is an exciting new field of research, with putative important diagnostic and pharmacological developments. Bioactive phytocompounds are being found to modulate AQPs including those expressed in the skin [120,121], and potent and isoform-specific synthetic compounds inhibiting AQPs begin to be available [30,122,123] with potentially novel therapeutic options dealing with the prognosis and cure of highly prevalent diseases. There are, therefore, good reasons to think that if, on the one hand, AQPs are involved in the destruction of the skin barrier in skin inflammatory diseases, on the other hand, their modulation through natural compounds or specific drugs could represent a new approach in preserving the integrity and the full functionality of the skin.

## Figures and Tables

**Figure 1 ijms-23-04020-f001:**
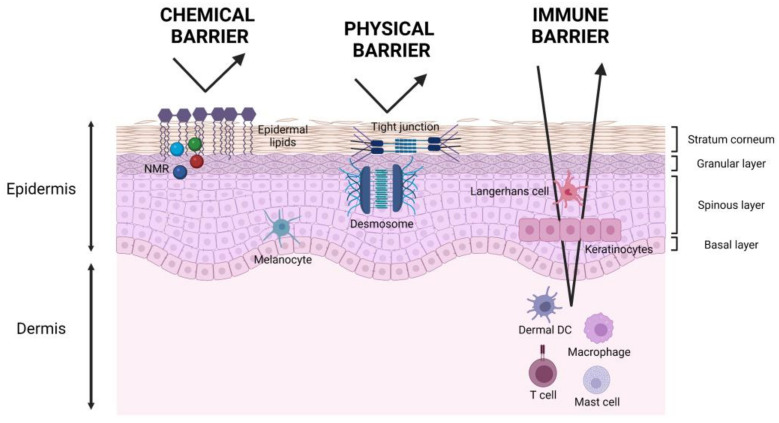
Schematic representation of skin barrier functions. The chemical barrier is mainly due to natural moisturizing factors (NMF) and epidermal lipids in the stratum corneum. The physical barrier is accomplished thanks to tight junctions and desmosomes in the granular and spinous layers. The immune barrier is carried on by skin-resident immune cells in the epidermis and dermis such as Langerhans cells and dermal dendritic cells, macrophages, resident T cells, and mast cells, respectively, and by epidermal keratinocytes.

**Figure 2 ijms-23-04020-f002:**
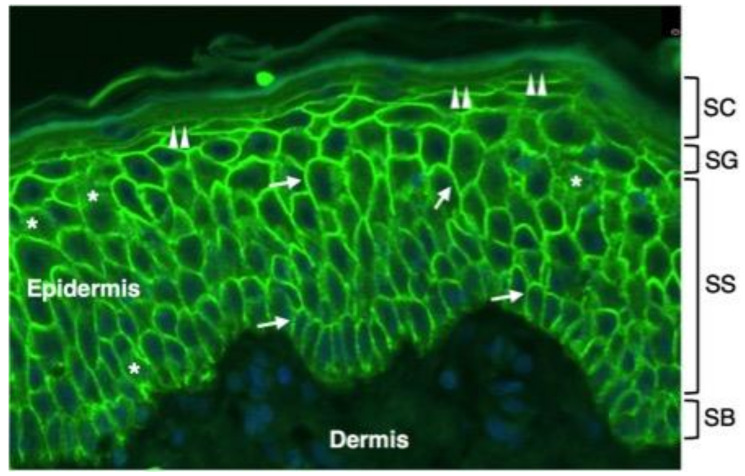
Immunofluorescence distribution of AQP3 in normal human epidermis. Shown is immunofluorescence analysis of AQP3 labelling in the epidermis of a normal human subject. Skin tissue was formalin-fixed and paraffin-embedded. Sections were stained using an anti-human AQP3 antibody recognizing AQP3 (Thermo Fisher Scientific, Monza, Italy). Cell nuclei were stained with DAPI (blue fluorescence). Strong AQP3 immunoreactivity (green fluorescence) is seen over the plasma membrane of stratum basale (SB) and stratum spinosum (SS) keratinocytes (arrows). Weak intracellular immunoreactivity is observed in the intracellular compartment (asterisks). The plasma membrane immunostaining of SG keratinocytes is lower than the one of the underlying epidermal layers (double arrowheads). No immunofluorescence is seen in the stratum corneum (SC) of the epidermis and in the dermis.

**Figure 3 ijms-23-04020-f003:**
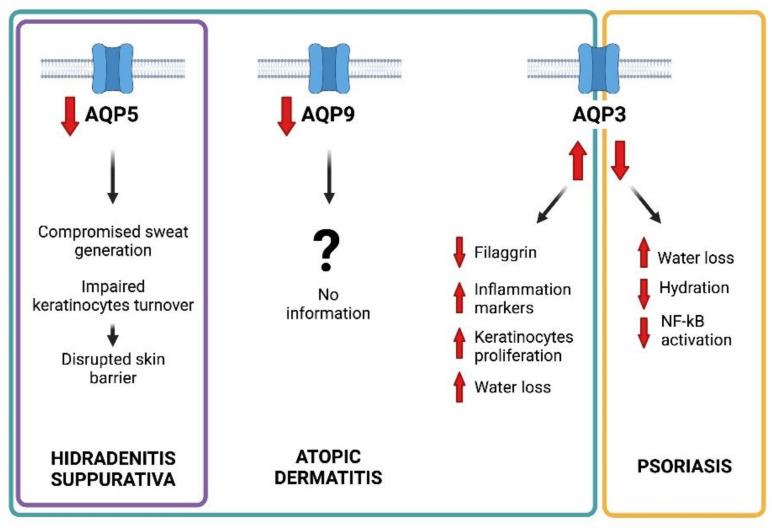
Graphical representation of the main pathophyioslogical roles suggested for AQP5, AQP9, and AQP3 in hidradenitis suppurativa, atopic dermatitis, and psoriasis, respectively.

**Table 1 ijms-23-04020-t001:** Localization, regulation and physiological roles of human/rodent skin aquaporins.

AQP	Permeability	Skin Layer, Cell Type and Subcellular Location	Agents/Conditions Regulating the Expression/Trafficking of the AQP	Suggested Roles in the Skin
AQP1	Water, hydrogen peroxide	Keratinocytes (ED)	Undefined	Creation of hydraulic pressure for cell migration
		Melanocytes (ED; SB)	Hypertonic stress ↑ (?)	Growth of melanocyte dendrites (?); melanosome transfer to keratinocytes (?)
		Fibroblast (D)	Hypertonic stress ↑	Cellular response to hypertonic stress (?)
		Vascular endothelial cells (D, HD)	Undefined	Water exchange blood-dermis (skin hydration)
AQP3	Water, glycerol, urea, hydrogen peroxide, ammonia	Keratinocytes (ED; SB: PM and IC; SS: PM and IC; SG: PM)	HDAC3 ↓; p73 ↑; PPAR*γ* ↑; LXR, RAR, RXR ↑; VD3 ↓; * high levels of extracellular Ca^2+^ ↓↑; osmotic stress (translocation to PM↑); circadian rhythm	Keratinocyte early differentiation **; keratinocyte proliferation and migration during wound healing; skin hydration (circadian rhythm); maintenance of epidermal water permeability barrier
		Dermal-resident T cells (PM)	Hapten-induced skin contact	Chemokine-dependent T cell migration in skin hypersensitivity to haptens
AQP5	Water, hydrogen peroxide	Sweat glands: secretory cells (APM, BPM); excretory duct cells (APM)	Muscarinic agonists	Sweat secretion
AQP7	Water, glycerol	Langherans cells (ED); Dermal dendritic cells (D)	Undefined	Cell migration, antigen uptake, immune surveillance
AQP9	Water, glycerol, hydrogen peroxide, ammonia	Keratinocytes (ED; upper SG)	VD3 ↑	Keratinocyte late differentiation; transcellular route for glycerol and urea movement (?)
AQP10	Water, glycerol	Keratinocytes (ED; SC)	Undefined	Barrier function

BPM, basolateral plasma membrane; D, dermis; ED, epidermis; HD, hypodermis; SB, stratum basale; SG, stratum granulosum; SS, stratum spinosum; SC, stratum corneum; PM, plasma membrane; APM, apical plasma membrane; IC, intracellular; VD3, 1,25-dihydroxyvitamin D3. *, decrease of non-glycosylated AQP3 and increase of glycosylated AQP3. **, controversial results in the literature. ↑, increase in AQP3 mRNA/protein expression or protein translocation to PM. ↓, decrease in AQP3 mRNA/protein expression.

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
