# Peer review of "Aquaporins Are One of the Critical Factors in the Disruption of the Skin Barrier in Inflammatory Skin Diseases"

_ijms, 2022, doi:10.3390/ijms23074020_

Round 1

Reviewer 1 Report

The role of AQP in skin homeostasis and inflammation is a novel topic of research, not completely understood. The manuscript is well written and highlights, effectively, previous works regarding AQP proteins in skin homeostasis, barrier function and skin inflammation. The cited references are well selected.
I have the following suggestion for the authors: 

the skin harbours a great number of T cells, in addition to innate cells. T cells are the main players during skin inflammation. Moreover, at least AQP3 and AQP9 are expressed in T cells. I suggest including T cell populations in figure 1, as part of the immune barrier, and also adding a brief description regarding the role of each AQP in T cells, and whether it has been explored in the skin.

Author Response

Review Report from Reviewer 1

Comments and Suggestions for Authors
The role of AQP in skin homeostasis and inflammation is a novel topic of research, not completely understood. The manuscript is well written and highlights, effectively, previous works regarding AQP proteins in skin homeostasis, barrier function and skin inflammation. The cited references are well selected.
I have the following suggestion for the authors:

 the skin harbours a great number of T cells, in addition to innate cells. T cells are the main players during skin inflammation. Moreover, at least AQP3 and AQP9 are expressed in T cells. I suggest including T cell populations in figure 1, as part of the immune barrier, and also adding a brief description regarding the role of each AQP in T cells, and whether it has been explored in the skin.

Answer. Figure 1 has been modified accordingly and T cells have been also included in Figure 1 legend. The reported expression of AQP3 in dermal-resident T cells has also been included to Table 1. A brief description regarding the role of the AQPs in T cells was also added.

We thank the Reviewer for the helpful criticism.

Reviewer 2 Report

The author reviews the role of aquaporins in skin keratinization, hydration, keratinocytes proliferation, water retention, barrier repair, wound healing, and immune response activation. Aquaporins-3 is slightly overrepresented in this manuscript, although it is the most abundant skin aquaglyceroporins. I would suggest the author:

  1. Re-edit a more appropriate title
  2. There is some repetition, especially in sections 4 and 5.
  3. Section 4.3: Some studies reported the relationship between AQP-5 and hyperhidrosis. Should summaries those studies in this section.
  4. Section 4.4: One study reported that AQP9 might involve in hapten-induced contact hypersensitivity. (PMID 26489517)
  5. Figure 3: may move to section 4.

Questions:

  1. Line 119-125: Which aquaporinsgroups should AQP5 be classified?
  2. What is your opinion: Are aquaporins involved in disrupting the skin barrier in inflammatory skin diseases?

Author Response

Review Report from Reviewer 2

Comments and Suggestions for Authors
The author reviews the role of aquaporins in skin keratinization, hydration, keratinocytes proliferation, water retention, barrier repair, wound healing, and immune response activation. Aquaporins-3 is slightly overrepresented in this manuscript, although it is the most abundant skin aquaglyceroporins. I would suggest the author:

Re-edit a more appropriate title

Answer. The title has been edited as requested to make it more informative.

There is some repetition, especially in sections 4 and 5.

Answer. We acknowledged the suggestion and removed the redundant text.

Section 4.3: Some studies reported the relationship between AQP-5 and hyperhidrosis. Should summaries those studies in this section.

Answer. The debated relationship between AQP5 and hyperhidrosis has been summarized in section 4.3.

 Section 4.4: One study reported that AQP9 might involve in hapten-induced contact hypersensitivity. (PMID 26489517)

Answer. The suggested involvement of neutrophil AQP9 in hapten-induced contact hypersensitivity has been now inserted in section 4.5

Figure 3: may move to section 4.

Answer. The figure 3 has been moved in section 5.

Questions:

Line 119-125: Which aquaporins groups should AQP5 be classified?

Answer. We thank the Reviewer for pointing out the oversight in question. We have now included AQP5 among the ortodox AQPs (section 3).

What is your opinion: Are aquaporins involved in disrupting the skin barrier in inflammatory skin diseases?

Answer. In the conclusive remarks we recalled the importance of AQPs in the skin barrier disruption in inflammatory skin diseases.

We thank the Reviewer for the helpful criticism.

Reviewer 3 Report

Review of manuscript

The well written manuscript is devoted to a problem of aquaporins involvement in the function of skin in selected inflammatory diseases of the skin. The manuscript comprises description of the skin barrier and its’ permeability as well as discussion devoted to aquaporins – their expression and physiological function. The following aquaporins were described AQP1, AQP3, AQP5, AQP7, AQP9, AQP10. The changes of aquaporins in the hidradenitis suppurativa, psoriasis and atopic dermatitis were discussed. The last part of the manuscript is comprised by conclusion that summarizes the most important information as well as contains therapeutic suggestions.

The manuscript could be interesting for the readers of International Journal of Molecular Sciences.

There are a few concerns with the study:

Major concerns

  1. The paper discusses the inflammatory diseases however the manuscript does not contain any definition of inflammation. Moreover short description of the inflammatory process would be beneficial for the study.
  2. The discussion related to inflammatory mediators (histamine, …) effect on the aquaporin expression and skin barrier would improve the manuscript.
  3. In the discussion devoted to full thickness wound healing is mainly focused on keratinocytes, however healing process is dependent also on other cells (myofibroblasts endothelial cells….) Especially wound contraction  is dependent on myofibroblasts. These information should be mentioned within the manuscript.

Minor concerns

  1. The Authors should consider replacement of the title of the final chapter from Conclusions into Summary and conclusions.

Author Response

Review report from Reviewer 3

Comments and Suggestions for Authors
The well written manuscript is devoted to a problem of aquaporins involvement in the function of skin in selected inflammatory diseases of the skin. The manuscript comprises description of the skin barrier and its’ permeability as well as discussion devoted to aquaporins – their expression and physiological function. The following aquaporins were described AQP1, AQP3, AQP5, AQP7, AQP9, AQP10. The changes of aquaporins in the hidradenitis suppurativa, psoriasis and atopic dermatitis were discussed. The last part of the manuscript is comprised by conclusion that summarizes the most important information as well as contains therapeutic suggestions.

The manuscript could be interesting for the readers of International Journal of Molecular Sciences.

There are a few concerns with the study:

Major concerns

The paper discusses the inflammatory diseases however the manuscript does not contain any definition of inflammation. Moreover short description of the inflammatory process would be beneficial for the study.

Answer. We thank the reviewer for the valuable suggestion, we added the induced inflammatory process for each described pathology.

The discussion related to inflammatory mediators (histamine, …) effect on the aquaporin expression and skin barrier would improve the manuscript.

Answer. What is known about the effect of inflammatory mediators on the aquaporin expression and skin barrier has now been discussed in several points of the manuscript also to answer the questions asked by Reviewer 1. We thank the Reviewer for the advice provided.

In the discussion devoted to full thickness wound healing is mainly focused on keratinocytes, however healing process is dependent also on other cells (myofibroblasts endothelial cells….) Especially wound contraction is dependent on myofibroblasts. These information should be mentioned within the manuscript.

Answer. We agree with the reviewer, the healing process is dependent on a large number of cells. However, our review was submitted in a special issue “Functional Defects of Keratinocytes in Inflammatory Skin Diseases”; for this reason we decided to focus our attention mostly on keratinocytes. In any case, the dependency of the healing process on other cells than keratinocytes has been briefly discussed (section 3.2).

Minor concerns

The Authors should consider replacement of the title of the final chapter from Conclusions into Summary and conclusions.

Answer. The title of section 6 has been edited as requested.

We thank the Reviewer for the helpful criticism.

Round 2

Reviewer 2 Report

The author has made a  significant modification to make the manuscript more complete; however, there is still one minor suggestion:

I would suggest using the phase of "one of the critical factors" rather than "key player" in the title.